# Age-related differences in driving behaviors among non-professional drivers in Egypt

**Ahmed Arafa, Lamiaa H. Saleh, Shaimaa A. Senosy**  *

Department of Public Health and Community Medicine, Beni-Suef University, Beni-Suef, Egypt

* shoshoahmed80@yahoo.com

## Abstract

### Purpose

This study aimed to investigate whether young and old non-professional drivers in South Egypt have aberrant driving behaviors compared with their middle-aged counterparts.

### Materials and methods

In this cross-sectional study, a total of 1764 non-professional drivers aged $\geq$ 19 years, residing in Beni-Suef in South Egypt, and having $\geq$ one year of driving experience were randomly selected. All drivers were asked to fill out a self-administered questionnaire, including personal information, driving habits, and the Arabic version of the Driver Behavior Questionnaire which evaluates driving violations, errors, and lapses.

### Results

This study included 560 young drivers (19–25 years), 850 middle-aged drivers (26–59 years), and 354 old drivers ($\geq$ 60 years). Compared with middle-aged drivers, young drivers reported more non-use of the seatbelt, eating while driving, and driving while feeling drowsy. Old drivers, in contrast, showed more careful driving behaviors including fewer violations, errors, and lapses and less likelihood of driving while feeling sleepy.

### Conclusion

This study supports the conception that young drivers pose less careful driving habits. Initiating educational programs targeting young drivers to improve their driving habits and create a traffic safety culture in Egypt is highly warranted.

## Introduction

With over 1.35 million annual fatalities worldwide, road traffic injuries (RTIs) represent the 8th leading cause of mortality for all ages. Recent trends indicated, however, that RTIs will be one of the top five causes of mortality by 2030 unless urgent actions are taken [1, 2]. In Egypt, RTIs contributed to 613 disability-adjusted life years lost per 100,000 population in 2016,

**Data Availability Statement:** All relevant data are within the manuscript and its Supporting Information files.

**Funding:** The author(s) received no specific funding for this work.

**Competing interests:** The authors have declared
that no competing interests exist.

making the country one of the most affected in Africa and the Middle East [3]. In this regard,
the United Nations aspired in its Agenda for Sustainable Development Goals to halve global
mortalities and morbidities from RTIs and enhance road safety for all through controlling the
risk factors for RTIs [4].

Human, environmental, road, and vehicle factors actively interact in RTI involvement [5–
7]. Human factors, in which aberrant driving behaviors are central, profoundly contribute to
RTIs [7, 8]. Aberrant driving behaviors can be categorized into violations, errors, and lapses.
While violations refer to intentional deviations from essential driving practices, errors refer to
incorrect driving practices associated with inadequate information. Different mechanisms pre-
sumably lead to different aberrant behaviors: violations result from risk-taking and errors
from low abilities; however, both behaviors significantly threaten road safety. Lapses encom-
pass occasional inattentions and memory failures and unlike violations and errors are less
likely to affect road safety [8, 9].

The issue of driving behaviors is multidimensional with various intersecting demographic,
social, psychological, legal, economic, and cultural factors [10, 11]. The age of drivers is among
the most influential factors affecting driving behaviors [9–16]. A cross-sectional study on 1600
drivers from the UK revealed a negative correlation between the age of drivers and their viola-
tions [9]. Alike, a study on 343 mini-bus drivers from Ethiopia showed that young age pre-
dicted risky driving behavior [7]. A study on 518 professional and non-professional drivers in
Egypt reported a higher prevalence of errors and lapses but not violations among drivers < 30
years, yet the significant association between age and driving behaviors disappeared after
adjustment for other sociodemographic features and driving distance [16].

The age of drivers can also cast the relationship between driving behaviors and RTIs as
young and old drivers could have a high risk of RTI involvement for different reasons [6, 9, 10,
12–16]. The increased probability of RTIs among young drivers can be a result of risky driving
behavior, a lack of driving skills and experience, and the propensity to drive in high-risk situa-
tions [10, 12, 13] while physical, sensorial, and cognitive age-related deteriorations that affect
driving ability could explain the high risk among old drivers [14, 15].

Since the prevalence rates of RTIs and aberrant driving behaviors in Egypt exceed those in
other Middle Eastern countries, Australia, and Western Europe [3, 16–20] and given the fact
that driving behavior is closely related to RTIs [8, 10], studying the potential associations
with aberrant driving behaviors is essential for a better prediction of RTIs in the country. Nev-
ertheless, data is lacking about the relationship between the age of drivers and their driving
behaviors. Detecting such a relationship can play a role in modifying the national driving regu-
lations and refining road safety awareness programs to suit the age of drivers. We, therefore,
conducted this cross-sectional study on a large sample of drivers in Beni-Suef city in Egypt to
investigate whether young or old drivers would show more aberrant driving behaviors com-
pared with middle-aged drivers.

## Subjects and methods

### Study design, population, and setting

This population-based, cross-sectional, analytical study was conducted on non-professional
drivers residing in Beni-Suef city, the Capital of Beni-Suef governorate in South Egypt, during
the period between October and December 2019. At the time of the 2017 census, more than
3.15 million people were residing in Beni-Suef governorate, and 30% of them were living in
Beni-Suef city [21]. Although Beni-Suef governorate has one of the fewest numbers of private
vehicles nationwide [21], it suffers a hefty burden of RTIs [16, 22]. The prevalence of RTI

involvement among non-professional drivers in the governorate during the past two years reached a high of 15.6% [16]. Beni-Suef governorate came on top of RTIs/total injuries list in Egypt with 51.2%, almost 20% higher than Ismailia governorate which came second and 35% higher than Cairo governorate [22].

In the current study, non-professional drivers were defined as people who were driving their vehicle or one owned by relatives or friends without being part- or full-time professional drivers. For the non-professional driver to be eligible for participation, he/she had to be aged ≥19 years and had been driving for at least one year.

## Sampling

The sample size was estimated using the Epi- Info version 7 StatCalc designed by the Centers for Disease Control and Prevention (CDC) and the World Health Organization (WHO). We more than doubled the least required sample size to enhance statistical power and avoid unpredicted low response rates. Eventually, we invited 2000 non-professional drivers to partic-ipate in this study.

To include a representative sample of non-professional drivers from Beni-Suef city, we divided the city areas according to their socioeconomic standards into three categories: high, middle, and low. Then, one area was randomly selected, by card withdrawal, from each socio-economic standard, and non-professional drivers residing in the selected areas were invited to participate in this study. A team of data collectors, led by one of the authors, visited the selected areas. Out of each area, 500 households or more were chosen using a random start, and data collectors moved from door to door asking those who were eligible to take part in the study. Those who gave their permission and signed their informed consent were handed a question-naire to fill out on their own before handing the questionnaire back to data collectors after one hour. No rewards were offered for participation.

## Data collection tool

Study participants were asked to fill out a self-administered questionnaire composed of two parts. The first part involved questions about age (years), sex (men or women), educational level (elementary or high), driving hours per day, the year model of vehicle, and some driving habits including eating and using the cellphone while driving, using the seatbelt, and driving while feeling sleepy or drowsy. Answer options for the driving habits were requested on a scale of "*never, seldom, sometimes, and always.*" The second part of the questionnaire included the Arabic version of the Driver Behavior Questionnaire (DBQ), which involved 26 items on a six-point scale from zero to five (*never, hardly ever, occasionally, quite often, frequently, and nearly all the time*). The items were divided as follows: ten items for violations such as "*disregarding the speed limits on a motorway,*" "*being involved in races,*" and "*sounding horn to annoy others*", eight items for errors such as "*applying sudden brakes,*" "*missing signs,*" and "*failing to recognize the pedestrians*", and eight items for lapses such as "*getting into wrong lanes,*" "*forgetting site of a car park,*" and "*misreading signs.*" High scores of DBQ indicated aberrant driving behavior [9, 23]. The factor analysis and reliability of the Arabic version of DBQ were first examined by Berner and colleagues who surveyed drivers from Arab Gulf countries [17]. The DBQ was also used to assess driving behaviors in Egypt, Australia, Sweden, the United Kingdom, Finland, and the Netherlands [16, 18–20]. However, since traffic culture varies between countries and given the absence of culture-specific items, it is hard to assume that drivers' perception of DBQ was the same across countries [17].

### Ethical consideration

The research proposal was approved by the Research Ethics Committee of the Faculty of Medicine Beni-Suef University, and the study was conducted in full accordance with the guidelines for the Declaration of Helsinki. Participants had to sign their informed consent forms before filling out their questionnaires.

### Statistical analyses

The Statistical Package for Social Science (SPSS) Version 22.0 (IBM SPSS Statistics for Windows, IBM Corporation, Armonk, New York) was used for data analysis. Age of drivers was categorized as follows: young drivers (19–25 years), middle-aged drivers (26–59 years), and old drivers ($\geq$ 60 years). Drivers were considered to have aberrant driving habits (eating or using the cellphone while driving, non-use of the seatbelt, and driving while feeling sleepy or drowsy) if they selected "*sometimes or always*" as answers for these habits. The chi-squared test was used to compare the sociodemographic characteristics of non-professional drivers among different age groups. Linear regression was used to detect the relationship between the age of drivers and their driving behaviors (violations, errors, and lapses). Logistic regression was used to detect the relationship between the age of drivers and their driving habits (eating, using the cellphone, and using the seatbelt while driving and driving while feeling sleepy or driving while feeling drowsy). Since our primary hypothesis suggested different forms of aberrant driving behaviors and habits among young and old drivers, we calculated betas (*B*s) for aberrant behaviors and odds ratios (ORs) for aberrant driving habits with their corresponding 95% confidence intervals (CIs) among young drivers (19–25 years) and old drivers ($\geq$ 60 years) in comparison with middle-aged drivers (26–59 years). Unadjusted and multivariable-adjusted linear and logistic regression models were conducted. The multivariable models included sex, education, vehicle model, and driving hours per day.

## Results

A total of 1764 non-professional drivers participated in this study with a response rate of 88.2%. Of them, 560 were young drivers (19–25 years), 850 were middle-aged drivers (26–60 years), and 354 were old drivers ($\geq$ 60 years). The group of old drivers included significantly more men, less educated people, and earlier vehicle models compared with the groups of middle-aged and young drivers (p<0.001) (Table 1).

After adjustment for sex, education, vehicle model, and driving hours per day, old drivers, compared with middle-aged drivers, reported fewer violations (*B* -1.70, 95% CI: -2.28, -1.12), errors (*B* -0.65, 95% CI: -1.05, -0.24), and lapses (*B* -0.81, 95% CI: -1.24, -0.37) (Table 2).

Regarding aberrant driving habits, young drivers were more likely to report non-use of the seatbelt (OR 1.56, 95% CI: 1.25, 1.95), eating while driving (OR 1.47, 95% CI: 1.15, 1.88), and driving while feeling drowsy (OR 1.75, 95% CI: 1.22, 2.52) compared with middle-aged drivers. In contrast, old drivers, compared with middle-aged ones, reported less driving while feeling sleepy (OR 0.49, 95% CI: 0.28, 0.84) and tended to show less use of the cellphone while driving (OR 0.77, 95% CI: 0.58, 1.02) and eating while driving (OR 0.74, 95% CI: 0.53, 1.03) (Table 3).

## Discussion

The current study illustrated the association of age with driving behaviors and habits among non-professional drivers in South Egypt. Young drivers (19–25 years) exhibited more aberrant driving habits including non-use of the seatbelt, eating while driving, and driving while feeling drowsy compared with middle-aged drivers (26–59 years). In contrast, old drivers ($\geq$ 60 years)

**Table 1. Sociodemographic characteristics of non-professional drivers distributed by age category.**

| Characteristics | | 19–25 years (n = 560) | 26–59 years (n = 850) | ≥ 60 years (n = 354) | P-value |
|---|---|---|---|---|---|
| **Age (Mean±Sd)** | | 21.43±2.01 | 38.79±8.87 | 64.38±3.79 | - - - |
| **Sex** | Men | 410 (73.2) | 633 (74.5) | 321 (90.7) | <0.001 |
| | Women | 150 (26.8) | 217 (25.5) | 33 (9.3) | |
| **Education** | Elementary | 194 (34.6) | 323 (38.0) | 220 (62.1) | <0.001 |
| | High | 366 (65.4) | 527 (62.0) | 134 (37.9) | |
| **Vehicle model** | Before 2000 | 64 (11.4) | 103 (12.1) | 75 (21.2) | <0.001 |
| | Recent | 496 (88.6) | 747 (87.9) | 279 (78.8) | |
| **Driving hours/day** | ≥4 | 279 (49.8) | 472 (55.5) | 193 (54.5) | 0.100 |
| | <4 | 281 (50.2) | 378 (44.5) | 161 (45.5) | |

showed more careful driving behaviors and habits than middle-aged drivers involving fewer violations, errors, and lapses and less likelihood of driving while feeling sleepy.

Our study came in line with several reports that showed aberrant driving behaviors and habits among young drivers. In one study, 504 young drivers (16–20 years) and 409 middle-aged drivers (25–45 years) in the United States were surveyed for their driving behaviors. The results showed that young drivers, compared with middle-aged drivers, did not just demonstrate more risky driving behaviors but lower risk perception of their driving behaviors as well [24]. Also, more than half of 484 Australian drivers (17–25 years) reported using the cellphone while driving. The young drivers were even unable to acknowledge the increased risk of RTIs associated with dialing, texting, and browsing while driving [25].

Young drivers are in the phase of seeking identity and building relationships with peers. While still evolving their identity and relationships, they tend to test their limits and abilities behind the wheel [26]. Therefore, young drivers demonstrate a high propensity for taking crash risks to fulfill their motives for experience-seeking, excitement, social influence, and prestige-seeking [12, 13]. Besides, young drivers who undertake or are exposed to risky driving behaviors have a low perception of driving risks [10].

**Table 2. *Betas* and 95% confidence intervals for non-professional drivers' aberrant driving behaviors according to their age category.**

| Driving behaviors | 19–25 years (n = 560) | 26–59 years (Ref) (n = 850) | ≥ 60 years (n = 354) |
|---|---|---|---|
| **Violations** | | | |
| **Mean (standard deviation)** | 15.8 (9.8) | 15.4 (9.2) | 12.1 (9.2) |
| **Unadjusted** | 0.39 (-0.62, 1.39) | 1 | -1.65 (-2.22, -1.08) |
| **Adjusted** | 0.64 (-0.35, 1.62) | 1 | -1.70 (-2.28, -1.12) |
| **Errors** | | | |
| **Mean (standard deviation)** | 12.8 (6.7) | 13.2 (6.5) | 11.6 (6.1) |
| **Unadjusted** | -0.41 (-1.11, 0.29) | 1 | -0.79 (-1.18, -0.39) |
| **Adjusted** | -0.31 (-1.00, 0.38) | 1 | -0.65 (-1.05, -0.24) |
| **Lapses** | | | |
| **Mean (standard deviation)** | 11.8 (7.2) | 12.8 (6.7) | 10.8 (7.0) |
| **Unadjusted** | -0.94 (-1.68, -0.20) | 1 | -0.99 (-1.41, -0.57) |
| **Adjusted** | -0.88 (-1.61, -0.15) | 1 | -0.81 (-1.24, -0.37) |

Adjusted for sex, education, vehicle model, and driving hours per day.

Linear regression was used.

**Table 3. Odds ratios and 95% confidence intervals for non-professional drivers' aberrant driving habits according to their age category.**

| Driving behaviors | 19–25 years (n = 560) | 26–59 years (Ref) (n = 850) | ≥ 60 years (n = 354) |
|---|---|---|---|
| **Not using the seatbelt (Sometimes or always)** | | | |
| **n (%)** | 338 (60.4) | 424 (49.9) | 187 (52.8) |
| **Unadjusted** | 1.53 (1.23, 1.90) | 1 | 1.13 (0.88, 1.44) |
| **Adjusted** | 1.56 (1.25, 1.95) | 1 | 0.90 (0.69, 1.17) |
| **Using the cellphone while driving (Sometimes or always)** | | | |
| **n (%)** | 205 (36.6) | 279 (32.8) | 94 (26.6) |
| **Unadjusted** | 1.18 (0.95, 1.48) | 1 | 0.74 (0.56, 0.98) |
| **Adjusted** | 1.20 (0.96, 1.51) | 1 | 0.77 (0.58, 1.02) |
| **Eating while driving (Sometimes or always)** | | | |
| **n (%)** | 164 (29.3) | 193 (22.7) | 60 (16.9) |
| **Unadjusted** | 1.41 (1.11, 1.80) | 1 | 0.70 (0.50, 0.96) |
| **Adjusted** | 1.47 (1.15, 1.88) | 1 | 0.74 (0.53, 1.03) |
| **Driving while feeling sleepy (Sometimes or always)** | | | |
| **n (%)** | 64 (11.4) | 86 (10.1) | 17 (4.8) |
| **Unadjusted** | 1.15 (0.81, 1.62) | 1 | 0.45 (0.26, 0.77) |
| **Adjusted** | 1.18 (0.84, 1.67) | 1 | 0.49 (0.28, 0.84) |
| **Driving while feeling drowsy (Sometimes or always)** | | | |
| **n (%)** | 68 (12.1) | 65 (7.6) | 24 (6.8) |
| **Unadjusted** | 1.67 (1.17, 2.39) | 1 | 0.88 (0.54, 1.43) |
| **Adjusted** | 1.75 (1.22, 2.52) | 1 | 0.96 (0.59, 1.59) |

Adjusted for sex, education, vehicle model, and driving hours per day.

Logistic regression was used.

Old drivers in this study reported more careful driving behaviors and habits. In agreement with our results, a study conducted on 1793 drivers (≥ 65 years) from the United States showed that 90.8% of drivers reported no violations during the past two years and the vast majority of old drivers strongly disapproved of the unsafe driving habits involving non-use of the seatbelt, use of the cellphone while driving, and driving while feeling drowsy. Adhering to safe driving habits was even more evident among drivers ≥ 75 years [27]. Another study analyzed 679 sleep-related vehicle crashes from police databases in England and showed that old drivers were less likely to be engaged in this kind of injury [28].

Unlike young drivers, old drivers are aware of their cognitive and physical limitations; therefore, they tend to exhibit self-monitoring behaviors that lead to a better risk estimation while driving. Also, the ability of safe driving is a process that takes a long time, and old drivers are experienced enough to avoid situations that might necessitate risky driving decisions [29–31]. Further, old drivers are not only concerned about their driving, but the driving of others as well. This concern was shown to be associated with careful driving habits, such as using the seatbelt [14].

Besides, it was suggested that the claim that old drivers are excessively involved in RTIs is affected by low mileage bias. Since old drivers typically drive less distance, they have a higher chance of RTI involvement per unit of distance compared with younger drivers who drive long distances. Thus, controlling for driving distance could attenuate the age-related differences in RTI involvement across age categories [32–34]. However, daily driving hours that reflect distance driven did not differ significantly across age categories in our study.

It should be noted that this study had some limitations. First, we included only non-professional drivers who carried different sociodemographic characteristics and driving behaviors compared with professional drivers [16]; therefore, the results of the current study cannot be generalized to all drivers. Second, this study covered one city in South Egypt. It could be speculated that drivers in different cities may not share the same driving behaviors. Third, driving behaviors were documented by self-reporting, making them vulnerable to recall bias and socially desirable response bias. However, a previous study showed that self-reporting of driving could be trusted, and the impact of socially desirable response in DBQ was minimal [35]. Fourth, the cross-sectional design of the study did not allow it to imply a causal relationship. Fifth, we did not collect data on RTI involvement to investigate the relationship between driving behaviors and RTIs. Asking people about RTI involvement would have discouraged many drivers who were involved in RTIs to participate in the study due to the fear of legal consequences leading to non-response bias.

In conclusion, this study indicated that young non-professional drivers in South Egypt exhibit aberrant driving habits. Hence, generating educational programs to improve driving behavior and create a traffic safety culture in Egypt is highly desirable. These programs should be tailored to suit young drivers.

## Supporting information

**S1 File. New driving.**
(SAV)

## Author Contributions

**Conceptualization:** Shaimaa A. Senosy.

**Formal analysis:** Shaimaa A. Senosy.

**Methodology:** Ahmed Arafa, Shaimaa A. Senosy.

**Supervision:** Shaimaa A. Senosy.

**Writing – original draft:** Ahmed Arafa, Shaimaa A. Senosy.

**Writing – review & editing:** Lamiaa H. Saleh, Shaimaa A. Senosy.

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
