## [Decision Letter · Decision Letter 0]

6 May 2020

PONE-D-20-06735

Age-related differences in driving behaviors among non-professional drivers in Egypt

PLOS ONE

Dear ASS.PROF senosy,

Thank you for submitting your manuscript to PLOS ONE. After careful consideration, we feel that it has merit but does not fully meet PLOS ONE’s publication criteria as it currently stands. Therefore, we invite you to submit a revised version of the manuscript that addresses the points raised during the review process.

Your paper has been reviewed by two acknowledged experts in the field covered by the manuscript. Overall, they highlight a certain potential in it, but many clarifications, amendments and improvements are needed to consider it for publication in PLOS ONE. Please refer to all the comments appended below for your guidance. Further, the English writing is considerably poor, and my recommendation is to submit the paper to a professional editorial proof-reading.

Also, the introduction of the paper lacks enough background and support for psychosocial risk factors that have been demonstrated to influence/affect profesional driver's risky road behaviors (i.e., stress, fatigue, etc.), and more context on it (even though you did not measure all these variables, but they still count) and discussion in the glance of your results is needed. For this stage, please consider to access original research papers published in high-impact journals, showing empirical data and (if possible) predictive models on the explanation of risky road behavior among Professional Drivers, even if not all the studies have been performed in Egypt.

We would appreciate receiving your revised manuscript by Jun 20 2020 11:59PM. To enhance the reproducibility of your results, we recommend that if applicable you deposit your laboratory protocols in protocols.io, where a protocol can be assigned its own identifier (DOI) such that it can be cited independently in the future. For instructions see: http://journals.plos.org/plosone/s/submission-guidelines#loc-laboratory-protocols

We look forward to receiving your revised manuscript.

Kind regards,

Sergio A. Useche, Ph.D.

Academic Editor

PLOS ONE

Additional Editor Comments:

Major language revisions are needed.

Journal Requirements:

Reviewers' comments:

Reviewer's Responses to Questions

**Comments to the Author**

1. Is the manuscript technically sound, and do the data support the conclusions?

Reviewer #1: No

Reviewer #2: Yes

2. Has the statistical analysis been performed appropriately and rigorously? 

Reviewer #1: Yes

Reviewer #2: Yes

3. Have the authors made all data underlying the findings in their manuscript fully available?

Reviewer #1: No

Reviewer #2: No

4. Is the manuscript presented in an intelligible fashion and written in standard English?

Reviewer #1: No

Reviewer #2: Yes

5. Review Comments to the Author

Reviewer #1: This manuscript presents a study of driver behavior by age group among non-professional drivers in an Egyptian city. I have several concerns with the manuscript.

1) The manuscript does not provide a strong rationale for conducting the study. It is already well-documented that young drivers are risk-takers, inexperienced, and sensation seekers and it is already well-documented that older drivers make mistakes while driving due to declining abilities and are not risk takers. Given that this paper focuses on a single city in Egypt, the manuscript should present arguments why it is important to investigate driver behavior in Beni-Suef.

2) The introduction does not show a thorough understanding of the constructs underlying the driving behavior questionnaire (DBQ). Yes, the questionnaire divides aberrant driving behaviors into violations, errors, and lapses, it does so because these behaviors presumably result from different mechanisms: Violations result from risk-taking and errors from declining abilities. Many studies do not even include the lapses construct because it is not clear what underlies this type of behavior. Given the focus on driver age in this study, the authors should generate hypotheses based on these mechanisms.

3) The DBQ driver behavior questionnaire has been used extensively around the world to study driver behavior by age group. This research should be reviewed briefly in the introduction not in the results section of the paper.

4) The description of the methods is missing critical details. How were specific drivers selected to be in the study? Were they compensated for participation? How was the questionnaire administered (e.g., paper and pencil; Internet)? How was the purported randomization achieved? Given the focus on driver age, why were the three age groups balanced for equal cell sizes? Why weren’t the groups also balanced by gender, since it is know that driver self-report is often influenced by gender?

5) The designation of “old driver” as being age 60 or older is counter to the contemporary research and thinking in the older driver literature. Work over the past decade defines an older driver as either 65 years or 70 years and older. Indeed, the most recent work has defined older driver as 75 years or older. The fact that the mean age of the older driver group in this study was less than 65 years of age indicates that this group would not be considered older drivers to other researchers in the field of aging and transportation. This also probably is the reason why no significant differences were found between middle and old drivers on overall errors, lapses, or violations.

6) The authors should justify why comparisons for young and old drivers were made for middle age drovers, but not between each other. One would expect the greatest differences when comparing young versus old.

7) The second paragraph of the discussion implies that aberrant and risky driving behaviors are distinct. I would argue that risky driving behavior is one type of aberrant driving behavior and would, in particular, fall under the violations category of the DBQ. The authors should clarify.

8) The discussion includes some statements that are not supported by the literature and/or do not include proper citations. For example, the fourth paragraph of the discussion states that old drivers are aware of their cognitive and physical limitations and they therefore tend to exhibit self-monitoring behavior. This statement of fact does not have a citation and it is not supported by the research on older driver self-regulation.

Reviewer #2: This study investigates differences in driving behaviour for younger and older drivers (compared to middle-aged drivers) in a large sample of Egyptian drivers. Young drivers report significantly more risk-taking and violations. Differences between middle and older drivers were less common with older drivers showing more cautious driving than middle aged.

The comparison groups of <25 and >59 are interesting. I was not sure that “middle-aged” was an appropriate term for the whole 25-59 age group, particularly for 26-39. However, I don’t have an immediate suggestion for a better label for this group.

Overall, I found this an interesting study, but believe that a deeper analysis, as outlined below, might substantially increase the impact of the work.

Abstract

Ors are shown for group comparisons, but odds ratios cannot be interpreted unless the scales of the variables involved is clear. Please clarify the scales. It looks like age is categorised whereas the DBQ scales are simply the item means or totals? This needs to be clarified. Standardising the DBQ scales might provide a more interpretable metric.

The OR for errors and lapses is written as if the difference between younger and middle-aged is significant but the CI on the OR includes 1. It needs to be clearer that this difference is ns, or perhaps left out of the abstract. This issue is also relevant in the discussion.

Introduction

P1 Older drivers are argued to be at risk of injuries due to cognitive decline. This is plausible, but there is work that provides an alternative explanation simply based on lower mileage. I recommend considering this paper in the consideration of this issue:

Langford, J., Methorst, R., & Hakamies-Blomqvist, L. (2006). Older drivers do not have a high crash risk—A replication of low mileage bias. Accident Analysis & Prevention, 38(3), 574-578. doi:https://doi.org/10.1016/j.aap.2005.12.002

Methods

Please say a little more about how the Arabic version of the DBQ was obtained and which of the many versions of the DBQ the questionnaire was based on. It looks like the commonly made distinction between aggressive and non-aggressive violations was not made here. On what basis was this decision made?

Analyses: Additional control for annual mileage would be helpful if that was recorded.

Results

As currently presented, the results are interesting, and with note of the point about metrics made regarding the abstract, can be interpreted.

I would find a more detailed analysis of the DBQ data more interesting.

I don’t think it can be assumed that the factor structure of the DBQ is stable across cultures. Therefore, the paper would benefit from discussion of this issue in the introduction and from analysis in the present paper. This might be an Exploratory or Confirmatory analysis, depending on whether the literature review provides a strong expectation for the factor structure to be observed in this context.

Was crash involvement measured? A key question about applying measures of driving risk developed in HICs to LMICs is whether they maintain their validity in their central purpose; to identify propensity for crash involvement. Therefore analyses testing the relationship with crash involvement would be helpful and the sample size seems big enough to provide a meaningful test of this relationship. Finding a significant relationship with crash involvement would support the application of the DBQ in Egypt, but if it is not found then that would still be an interesting result for the research community. Coverage of this issue in the introduction would also be helpful. This needs to be raised as a limitation if crash involvement was not measured.

Discussion

Ensure the discussion focusses only on significant comparisons.

The idea of self-monitoring in older drivers is interesting and plausible. However, I was unclear how this was supported by the data collected.

6. PLOS authors have the option to publish the peer review history of their article (what does this mean?). If published, this will include your full peer review and any attached files.

Reviewer #1: No

Reviewer #2: No

---

## [Author Response · Author response to Decision Letter 0]

11 May 2020

Dear Editor in Chief,

Thank you for the valuable comments.

Herein, we included our response to the comments made by the editor and reviewers. 

Editor

1. Further, the English writing is considerably poor, and my recommendation is to submit the paper to a professional editorial proof-reading.

Response: The manuscript was reviewed by a professional editorial proof-reading as suggested. 

2. Also, the introduction of the paper lacks enough background and support for psychosocial risk factors that have been demonstrated to influence/affect profesional driver's risky road behaviors (i.e., stress, fatigue, etc.), and more context on it (even though you did not measure all these variables, but they still count) and discussion in the glance of your results is needed. For this stage, please consider to access original research papers published in high-impact journals, showing empirical data and (if possible) predictive models on the explanation of risky road behavior among Professional Drivers, even if not all the studies have been performed in Egypt.

Response: The introduction was modified as suggested (Page 5,6). We added brief descriptions for studies assessing the relationship between the age of drivers and their driving behaviors and crash involvement. We also added a brief description of the factors affecting driving behaviors (Page 5 lines 19-25 & Page 6 lines 26-36). 

Journal Requirements

 Response: Modified throughout the manuscript as requested. 

 Response: The ethical considerations section was modified as requested (Page 9 lines 103-107). 

Response: We added more details about the questionnaire (Page 8 lines 82-98 & Page 9 lines 99-102). We also made all data available. 

Reviewers' comments

Reviewer's Responses to Questions

Comments to the Author

 1. Is the manuscript technically sound, and do the data support the conclusions?

Reviewer #1: No

Reviewer #2: Yes

Response: We conducted rigorous modifications to make the manuscript technically clear. 

2. Has the statistical analysis been performed appropriately and rigorously?

Reviewer #1: Yes

Reviewer #2: Yes

 Response: Thank you.

3. Have the authors made all data underlying the findings in their manuscript fully available?

Reviewer #1: No

Reviewer #2: No

Response: We made all data available.

4. Is the manuscript presented in an intelligible fashion and written in standard English?

Reviewer #1: No

Reviewer #2: Yes

Response: The manuscript was reviewed by a professional editorial proof-reading. 

Review Comments to the Author

Reviewer #1: This manuscript presents a study of driver behavior by age group among non-professional drivers in an Egyptian city. I have several concerns with the manuscript.

1) The manuscript does not provide a strong rationale for conducting the study. It is already well-documented that young drivers are risk-takers, inexperienced, and sensation seekers and it is already well-documented that older drivers make mistakes while driving due to declining abilities and are not risk takers. Given that this paper focuses on a single city in Egypt, the manuscript should present arguments why it is important to investigate driver behavior in Beni-Suef.

Response: We added an argument about the importance of investigating the behaviors of drivers in Egypt (Page 6 lines 37-46). We made a further argument about why studying the associations with driving behaviors was required in Beni-Suef governorate (Page 7 lines 54-61).

2) The introduction does not show a thorough understanding of the constructs underlying the driving behavior questionnaire (DBQ). Yes, the questionnaire divides aberrant driving behaviors into violations, errors, and lapses, it does so because these behaviors presumably result from different mechanisms: Violations result from risk-taking and errors from declining abilities. Many studies do not even include the lapses construct because it is not clear what underlies this type of behavior. Given the focus on driver age in this study, the authors should generate hypotheses based on these mechanisms.

Response: We agree with you, therefore, we added further descriptions of driving behaviors and their possible associations with age. We also added brief examples of previous studies in this regard. As suggested, we generated a hypothesis postulating that young drivers could have more violations and errors while old drivers could have more lapses (Page 5 lines 19-25 & Page 6 lines 26-36). 

3) The DBQ driver behavior questionnaire has been used extensively around the world to study driver behavior by age group. This research should be reviewed briefly in the introduction not in the results section of the paper.

Response: We added a brief review of earlier research investigating the relationship between the age of drivers and their driving behaviors to the introduction section (Page 5 lines 19-25 & Page 6 lines 26-36).

4) The description of the methods is missing critical details. How were specific drivers selected to be in the study? Were they compensated for participation? How was the questionnaire administered (e.g., paper and pencil; Internet)? How was the purported randomization achieved? Given the focus on driver age, why were the three age groups balanced for equal cell sizes? Why weren’t the groups also balanced by gender, since it is know that driver self-report is often influenced by gender?

Response: We added more details to the methods section about the selection and data collection process (Page 8 lines 76-81). 

Regarding the possibility of balancing the group by sex, this study was cross-sectional, and the cut-offs of age groups were decided before data collection, thus, drivers were not equally distributed across age groups: 560 young, 850 middle-aged, and 354 old. Besides, because women represent a small portion of non-professional drivers in Egypt and putting in mind that seeing a woman driving in South Egypt 30 years ago was uncommon, it was predicted to have a small number of women in the old age group (≥ 60 years) of drivers. However, to minimize the effect of sex, we adjusted the differences for sex and other variables, and it could be shown in (Table 2) that the adjusted model did not differ significantly from the unadjusted one.

 5) The designation of “old driver” as being age 60 or older is counter to the contemporary research and thinking in the older driver literature. Work over the past decade defines an older driver as either 65 years or 70 years and older. Indeed, the most recent work has defined older driver as 75 years or older. The fact that the mean age of the older driver group in this study was less than 65 years of age indicates that this group would not be considered older drivers to other researchers in the field of aging and transportation. This also probably is the reason why no significant differences were found between middle and old drivers on overall errors, lapses, or violations.

Response: While most studies from the USA, Europe, and Japan use 65 years as a cut-off for old age, the Egyptian studies use 60 years as a cut-off. This cut-off is widely used nationwide for many reasons; 1- It is the age of pension in Egypt, 2- It is the age used by the Egyptian Census to refer to old people, 3- It is the age used by the Government to offer social welfare programs for old people, and 4- The overall average of life expectancy in Egypt is 71.5 years compared with 84.5 years in Japan, 83.5 years in Italy, Spain, Switzerland, and Australia, and 79 years in the USA. 

• Aly HY, Hamed AF, Mohammed NA. Depression among the elderly population in Sohag governorate. Saudi Med J. 2018;39(2):185‐190. 

• Ahmed D, El Shair IH, Taher E, Zyada F. Prevalence and predictors of depression and anxiety among the elderly population living in geriatric homes in Cairo, Egypt. J Egypt Public Health Assoc. 2014 Dec;89(3):127-35.

• Central Agency for Public Mobilization and Statistics (CAPMAS). Egypt statistics. Final results of 2017 Census; http://www.capmas.gov.eg

6) The authors should justify why comparisons for young and old drivers were made for middle age drovers, but not between each other. One would expect the greatest differences when comparing young versus old.

Response: Since our primary hypothesis was to detect aberrant driving behaviors in young and old drivers and to test whether young drivers had more violations and errors and old drivers had more lapses, we had to use the middle-aged people as a reference group because in our situation both young and old groups were study groups. 

7) The second paragraph of the discussion implies that aberrant and risky driving behaviors are distinct. I would argue that risky driving behavior is one type of aberrant driving behavior and would, in particular, fall under the violations category of the DBQ. The authors should clarify.

Response: We agree with you and we modified this section as suggested (Page 11 line 151).

8) The discussion includes some statements that are not supported by the literature and/or do not include proper citations. For example, the fourth paragraph of the discussion states that old drivers are aware of their cognitive and physical limitations and they therefore tend to exhibit self-monitoring behavior. This statement of fact does not have a citation and it is not supported by the research on older driver self-regulation.

Response: We clarified the citation and we added 2 more citations suggesting self-monitoring behavior among old drivers (Page 12 lines 176-182). However, we agree with you that some sections need proper citations and therefore we updated and added some references. 

• Anstey K, Wood J, Lord S, Walker J. Cognitive, sensory and physical factors enabling driving safety in older adults. Clin Psychol Rev, 2005; 25:45-65.

• Conlon E, Rahaley N, Davis J. The influence of age-related health difficulties and attitudes toward driving on driving self-regulation in the baby boomer and older adult generations. Accid Anal Prev, 2017; 102:12-22.

• Molnar L, Eby D, Charlton J, Langford J, Koppel S, Marshall S, et al. Driving avoidance by older adults: is it always self-regulation? Accid Anal Prev, 2013; 57:96-104. 

Reviewer #2: This study investigates differences in driving behaviour for younger and older drivers (compared to middle-aged drivers) in a large sample of Egyptian drivers. Young drivers report significantly more risk-taking and violations. Differences between middle and older drivers were less common with older drivers showing more cautious driving than middle aged.

1). The comparison groups of <25 and >59 are interesting. I was not sure that “middle-aged” was an appropriate term for the whole 25-59 age group, particularly for 26-39. However, I don’t have an immediate suggestion for a better label for this group.

Response: Defining people <60 years as middle-aged was used before in research.

• Tsugane S, Sasaki S, Tsubono Y. Under- and overweight impact on mortality among middle-aged Japanese men and women: a 10-y follow-up of JPHC study cohort I. Int J Obes Relat Metab Disord. 2002 Apr;26(4):529-37.

• Yoshida M, Inoue M, Iwasaki M, Tsugane S; JPHC Study Group. Association of body mass index with risk of age-related cataracts in a middle-aged Japanese population: the JPHC Study. Environ Health Prev Med. 2010 Nov;15(6):367-73.

Overall, I found this an interesting study, but believe that a deeper analysis, as outlined below, might substantially increase the impact of the work.

Abstract

2). Ors are shown for group comparisons, but odds ratios cannot be interpreted unless the scales of the variables involved is clear. Please clarify the scales. It looks like age is categorised whereas the DBQ scales are simply the item means or totals? This needs to be clarified. Standardising the DBQ scales might provide a more interpretable metric.

Response: Thank you for this comment. We clarified the cut-offs for violations, errors, and lapses in the abstract section as suggested (Page 2). 

3). The OR for errors and lapses is written as if the difference between younger and middle-aged is significant but the CI on the OR includes 1. It needs to be clearer that this difference is ns, or perhaps left out of the abstract. This issue is also relevant in the discussion.

Response: We rephrased the abstract to differentiate between the absolute statistically significant associations and the tendency of some factors to be associated with aberrant driving behaviors (Page 2).

Introduction

4). P1 Older drivers are argued to be at risk of injuries due to cognitive decline. This is plausible, but there is work that provides an alternative explanation simply based on lower mileage. I recommend considering this paper in the consideration of this issue:

Langford, J., Methorst, R., & Hakamies-Blomqvist, L. (2006). Older drivers do not have a high crash risk—A replication of low mileage bias. Accident Analysis & Prevention, 38(3), 574-578. doi:https://doi.org/10.1016/j.aap.2005.12.002

Response: Thank you. This study was very helpful. We added this explanation to the discussion part to justify our findings (Page 12 lines 183-189).

Methods

5). Please say a little more about how the Arabic version of the DBQ was obtained and which of the many versions of the DBQ the questionnaire was based on. It looks like the commonly made distinction between aggressive and non-aggressive violations was not made here. On what basis was this decision made?

Response: More descriptions were added to the Arabic version of DBQ (Page 8 lines 82-98 & Page 9 lines 99-102). We made this decision based on the low scores of reliability and factorial analyses shown in a previous study using the Arabic version of DBQ and dividing the behaviors into four aspects.

• Bener A, Ozkan T, Lajunen T. The driver behaviour questionnaire in Arab Gulf Countries: Qatar and United Arab Emirates. Accid Anal Prev, 2008; 40:1411-1417.

6). Analyses: Additional control for annual mileage would be helpful if that was recorded.

Response: Since we limited our study population to nonprofessional drivers and given the fact that the streets in Egypt, including Beni-Suef, are very crowded, we thought that using driving hours may be more realistic in the current situation that mileage. Thus, we adjusted our results for driving hours.

Results

7). As currently presented, the results are interesting, and with note of the point about metrics made regarding the abstract, can be interpreted.

Response: Thank you. The abstract was modified as suggested in the first comment.

8). I would find a more detailed analysis of the DBQ data more interesting. I don’t think it can be assumed that the factor structure of the DBQ is stable across cultures. Therefore, the paper would benefit from discussion of this issue in the introduction and from analysis in the present paper. This might be an Exploratory or Confirmatory analysis, depending on whether the literature review provides a strong expectation for the factor structure to be observed in this context.

Response: We further discussed the possible cultural differences across DBQ versions (Page 8 lines 82-98 & Page 9 lines 99-102). 

9). Was crash involvement measured? A key question about applying measures of driving risk developed in HICs to LMICs is whether they maintain their validity in their central purpose; to identify propensity for crash involvement. Therefore analyses testing the relationship with crash involvement would be helpful and the sample size seems big enough to provide a meaningful test of this relationship. Finding a significant relationship with crash involvement would support the application of the DBQ in Egypt, but if it is not found then that would still be an interesting result for the research community. Coverage of this issue in the introduction would also be helpful. This needs to be raised as a limitation if crash involvement was not measured.

Response: We did not assess crash involvement for 2 reasons: 1- This would greatly undermine the response rate leading to a high possibility of non-response bias because most traffic crashes that do not include injuries or include minor injuries are not reported to the police, so participants would be afraid of getting in legal troubles. A recent study from Egypt on crash involvement showed a response rate of only 52% and non-respondents expressed their concerns regarding data security. 2- Since many traffic crashes are not reported, we would not have any method of verification. However, we added this point as a limitation (Page 13 lines 199-202). 

• Arafa A, El-Setouhy M, Hirshon J. Driving behavior and road traffic crashes among professional and nonprofessional drivers in South Egypt. Int J Inj Contr Safety Promot, 2019; 26:372-378.

Discussion

10). Ensure the discussion focusses only on significant comparisons. The idea of self-monitoring in older drivers is interesting and plausible. However, I was unclear how this was supported by the data collected.

 Response: We limited the discussion to statistically significant associations as suggested (Page 10 lines 145-147 & Page 11 lines 148-150). Unfortunately, we did not assess self-monitoring, but previous studies suggested this explanation to justify the relatively safe driving of old people. Therefore, we cited these studies to explain our findings (Page 12 lines 176-182).

• Anstey K, Wood J, Lord S, Walker J. Cognitive, sensory and physical factors enabling driving safety in older adults. Clin Psychol Rev, 2005; 25:45-65. 

• Conlon E, Rahaley N, Davis J. The influence of age-related health difficulties and attitudes toward driving on driving self-regulation in the baby boomer and older adult generations. Accid Anal Prev, 2017; 102:12-22.

• Molnar L, Eby D, Charlton J, Langford J, Koppel S, Marshall S, et al. Driving avoidance by older adults: is it always self-regulation? Accid Anal Prev, 2013; 57:96-104.

---

## [Decision Letter · Decision Letter 1]

6 Jul 2020

PONE-D-20-06735R1

Age-related differences in driving behaviors among non-professional drivers in Egypt

PLOS ONE

Dear Dr. senosy,

Thank you for submitting your manuscript to PLOS ONE. After careful consideration, we feel that it has merit but does not fully meet PLOS ONE’s publication criteria as it currently stands. Therefore, we invite you to submit a revised version of the manuscript that addresses the points raised during the review process.

The Reviewer # 2 has provided additional comments that I found valuable and may contribute to improve the paper. Please address them carefully during your revisions of the paper. Also, and apart from the comments appended by the reviewer, I would like the authors could improve the literature review, in which much is discussed on some behavioral contributors of non-professional drivers, but the task-related issues of professional drivers, i.e., providing context on why it was important to differentiate them are (comparatively) uncompensated.

We look forward to receiving your revised manuscript.

Kind regards,

Sergio A. Useche, Ph.D.

Academic Editor

PLOS ONE

Reviewers' comments:

Reviewer's Responses to Questions

**Comments to the Author**

1. If the authors have adequately addressed your comments raised in a previous round of review and you feel that this manuscript is now acceptable for publication, you may indicate that here to bypass the “Comments to the Author” section, enter your conflict of interest statement in the “Confidential to Editor” section, and submit your "Accept" recommendation.

Reviewer #2: All comments have been addressed

2. Is the manuscript technically sound, and do the data support the conclusions?

Reviewer #2: Partly

3. Has the statistical analysis been performed appropriately and rigorously? 

Reviewer #2: Yes

4. Have the authors made all data underlying the findings in their manuscript fully available?

Reviewer #2: Yes

5. Is the manuscript presented in an intelligible fashion and written in standard English?

Reviewer #2: Yes

6. Review Comments to the Author

Reviewer #2: Many thanks to the authors for their thoughtful responses to my comments. I remain of the opinion that the study provides results that will interesting to the research community. I have re-read the ms and provide further comments below.

Abstract

I am not convinced the sampling is random - were these volunteers? If so, volunteering is not random, so this needs to be clarified.

Results- I still find the ORs difficult to interpret. I recommend removing the Ors from the abstract an just describing the significant associations in the data. I do not believe any interpretation should be given for non-significant associations, anywhere in the ms.

Introduction

Now provides a solid introduction to the work in my reading.

Method

I don’t see the authors have dichotomised the DBQ scales to identify aberrant driving. The DBQ provides continuous measures of driving constructs and dichotomising them reduces the power of analysis without any benefit so far as I can see. I recommend analyses are re-run using the continuousDBQ scores as the outcome variables. This may mean that some of the tendencies identified in the data but falling below significance will become significant.

Discussion

No interpretation or consideration should be given to ns results.

P12Line 176. The point about self-monitoring older drivers needs to be grounded in literature.

7. PLOS authors have the option to publish the peer review history of their article (what does this mean?). If published, this will include your full peer review and any attached files.

Reviewer #2: **Yes: **Richard Rowe

---

## [Author Response · Author response to Decision Letter 1]

8 Jul 2020

Dear editor-in-chief,

Thank you for the favorable response.

Herein, our response to comments of reviewer 2

1. Abstract

1.1. I am not convinced the sampling is random - were these volunteers? If so, volunteering is not random, so this needs to be clarified.

Response: Thank you for this comment. The sample was randomly selected. The selection process was described in detail (lines 72-81). Since data were self-reported and given the fact that we offered no incentives for participation, it could be speculated that different forms of bias could be present, and we already cited these limitations (lines 191-194). 

1.2. Results- I still find the ORs difficult to interpret. I recommend removing the Ors from the abstract an just describing the significant associations in the data. I do not believe any interpretation should be given for non-significant associations, anywhere in the ms.

Response: We modified the results in the abstract section as suggested (page 3).

2. Introduction

2.1. Now provides a solid introduction to the work in my reading.

Response: Thank you.

3. Method

3.1. I don’t see the authors have dichotomised the DBQ scales to identify aberrant driving. The DBQ provides continuous measures of driving constructs and dichotomising them reduces the power of analysis without any benefit so far as I can see. I recommend analyses are re-run using the continuous DBQ scores as the outcome variables. This may mean that some of the tendencies identified in the data but falling below significance will become significant.

Response: While we appreciate your opinion, we have some concerns regarding treating DBQs scales as continuous scales rather than dichotomizing them for the following reasons:

1. This study aims, in the first place, to draw policies for traffic measures in South Egypt, therefore, the emerging results will be presented to officials and academics who might carry limited or no expertise in statistical analyses. We believe that presenting the results in the form of categorical variables will be easier to understand and can deliver our message directly. It is more convenient to claim that the younger age may be associated with a 46% increase in the risk of driving violations compared with the middle-aged than claiming that transition from middle age category to young age category may lead to an increase by 0.39 units on the violation scale of driving behaviors.

2. In response to your comment, we already re-ran the analysis, however, the results did not materially change.

3. In case of conducting linear regression to compute beta coefficients and their 95% CIs for violations, errors, and lapses, we will face another problem that other driving habits (using seatbelts, cellphones, eating, driving while feeling drowsy, and driving while feeling sleepy) are not continuous variables and conducting logistic regression in their case is a must. So, in such a case, the upper half of the table will be conducted by linear regression and presented by unit change and the lower half of the table will be conducted by logistic regression and presented by odds. Also, the descriptive data in the upper half will be presented by mean and Sd or median and IQR while the lower half will be presented by frequencies and percentages. We think that this inconsistency may leave the reader confused. 

4. Discussion

4.1. No interpretation or consideration should be given to ns results.

Response: We modified this point as suggested.

4.2. P12Line 176. The point about self-monitoring older drivers needs to be grounded in literature.

Response: This part is cited in references 29, 30, and 3

---

## [Decision Letter · Decision Letter 2]

11 Aug 2020

PONE-D-20-06735R2

Age-related differences in driving behaviors among non-professional drivers in Egypt

PLOS ONE

Dear Dr. senosy,

Thank you for submitting your manuscript to PLOS ONE. After careful consideration, we feel that it has merit but does not fully meet PLOS ONE’s publication criteria as it currently stands. Therefore, we invite you to submit a revised version of the manuscript that addresses the points raised during the review process.

Your revised manuscript has been reviewed again, raising new comments that need your attention, and I append below for your guidance. Please try to address them as rigorously as possible, in order to make a prompt editorial decision once you will resubmit the paper.

As a personal recommendation, please try to support the technical assumptions of the analyses performed (BQ paradigm) with more pertinent literature on it.

We look forward to receiving your revised manuscript.

Kind regards,

Sergio A. Useche, Ph.D.

Academic Editor

PLOS ONE

Reviewers' comments:

Reviewer's Responses to Questions

**Comments to the Author**

1. If the authors have adequately addressed your comments raised in a previous round of review and you feel that this manuscript is now acceptable for publication, you may indicate that here to bypass the “Comments to the Author” section, enter your conflict of interest statement in the “Confidential to Editor” section, and submit your "Accept" recommendation.

Reviewer #2: (No Response)

2. Is the manuscript technically sound, and do the data support the conclusions?

Reviewer #2: Yes

3. Has the statistical analysis been performed appropriately and rigorously? 

Reviewer #2: No

4. Have the authors made all data underlying the findings in their manuscript fully available?

Reviewer #2: Yes

5. Is the manuscript presented in an intelligible fashion and written in standard English?

Reviewer #2: Yes

6. Review Comments to the Author

Reviewer #2: Many thanks for your responses. Most are very helpful, but I do believe the ms will be much more valuable if the DBQ is treated as a continuous measure.

I can see the advantages of dichotomising the DBQ scales for ease of communication to non-academic audiences. However, it is an oversimplification that may mislead. There is no evidence that the cutpoints you have identified are in anyway meaningful in terms of risk of crash, but if you present the results in this way, any audience, particularly an uninformed one, is likely to think that they do mean something.

The audience you are writing for at PLOS ONE are academics who are familiar with interpreting continuous statistics in general. The audience with expertise in driving behaviour will expect that the DBQ will be treated as a continuous measure as it has always been in the very many studies that I have authored and read using this instrument.

The problem of some analyses addressing dichotomous variables and some addressing continuous is a common one in writing research papers, so I have no doubt you will be able to find an unambiguous way to present your tables.

7. PLOS authors have the option to publish the peer review history of their article (what does this mean?). If published, this will include your full peer review and any attached files.

Reviewer #2: No

---

## [Author Response · Author response to Decision Letter 2]

12 Aug 2020

Reviewer #2: Many thanks for your responses. Most are very helpful, but I do believe the ms will be much more valuable if the DBQ is treated as a continuous measure.

I can see the advantages of dichotomising the DBQ scales for ease of communication to non-academic audiences. However, it is an oversimplification that may mislead. There is no evidence that the cutpoints you have identified are in anyway meaningful in terms of risk of crash, but if you present the results in this way, any audience, particularly an uninformed one, is likely to think that they do mean something.

The audience you are writing for at PLOS ONE are academics who are familiar with interpreting continuous statistics in general. The audience with expertise in driving behaviour will expect that the DBQ will be treated as a continuous measure as it has always been in the very many studies that I have authored and read using this instrument.

The problem of some analyses addressing dichotomous variables and some addressing continuous is a common one in writing research papers, so I have no doubt you will be able to find an unambiguous way to present your tables.

Response: 

Thank you for your informative comment. We treated DBQ scales as continuous data as suggested and made some changes accordingly.

1- We divided table 2 to tables 2 and 3: table 2 for violence, errors, and lapses with associations conducted using linear regression and table 3 for driving habits such as eating, using the cell phone, and using the seatbelt during driving with associations conducted using logistic regression.

2- Some changes were made accordingly in the methods (lines 106-123), results (lines 130-140), and discussion sections (142-148).

---

## [Decision Letter · Decision Letter 3]

19 Aug 2020

Age-related differences in driving behaviors among non-professional drivers in Egypt

PONE-D-20-06735R3

Dear Dr. senosy,

We’re pleased to inform you that your manuscript has been judged scientifically suitable for publication and will be formally accepted for publication once it meets all outstanding technical requirements.

Kind regards,

Sergio A. Useche, Ph.D.

Academic Editor

PLOS ONE

Additional Editor Comments (optional):

Reviewers' comments:

Reviewer's Responses to Questions

**Comments to the Author**

1. If the authors have adequately addressed your comments raised in a previous round of review and you feel that this manuscript is now acceptable for publication, you may indicate that here to bypass the “Comments to the Author” section, enter your conflict of interest statement in the “Confidential to Editor” section, and submit your "Accept" recommendation.

Reviewer #2: All comments have been addressed

2. Is the manuscript technically sound, and do the data support the conclusions?

Reviewer #2: Yes

3. Has the statistical analysis been performed appropriately and rigorously? 

Reviewer #2: Yes

4. Have the authors made all data underlying the findings in their manuscript fully available?

Reviewer #2: Yes

5. Is the manuscript presented in an intelligible fashion and written in standard English?

Reviewer #2: Yes

6. Review Comments to the Author

Reviewer #2: Many thanks for responding carefully to my comments. My only final suggestions are:

Intro line 1 fatalities not mortalities

Line 118 betas not beats

7. PLOS authors have the option to publish the peer review history of their article (what does this mean?). If published, this will include your full peer review and any attached files.

Reviewer #2: **Yes: **Richard Rowe

---

## [Editor Report · Acceptance letter]

21 Aug 2020

PONE-D-20-06735R3 

Age-related differences in driving behaviors among non-professional drivers in Egypt 

Dear Dr. Senosy:

I'm pleased to inform you that your manuscript has been deemed suitable for publication in PLOS ONE. Congratulations! Your manuscript is now with our production department. 

Kind regards, 

on behalf of

Dr. Sergio A. Useche 

Academic Editor

PLOS ONE